# Anticancer Nanotherapeutics in Clinical Trials: The Work behind Clinical Translation of Nanomedicine

**DOI:** 10.3390/ijms232113368

**Published:** 2022-11-02

**Authors:** Alessandro Parodi, Ekaterina P. Kolesova, Maya V. Voronina, Anastasia S. Frolova, Dmitry Kostyushev, Daria B. Trushina, Roman Akasov, Tatiana Pallaeva, Andrey A. Zamyatnin

**Affiliations:** 1Scientific Center for Translation Medicine, Sirius University of Science and Technology, 354340 Sochi, Russia; 2Institute of Molecular Medicine, Sechenov First Moscow State Medical University, 119991 Moscow, Russia; 3Martsinovsky Institute of Medical Parasitology, Tropical and Vector-Borne Diseases, Sechenov First Moscow State Medical University, 119991 Moscow, Russia; 4Institute of Molecular Theranostics, Sechenov First Moscow State Medical University, 119991 Moscow, Russia; 5Federal Scientific Research Center «Crystallography and Photonics», Russian Academy of Sciences, 119333 Moscow, Russia; 6Shemyakin-Ovchinnikov Institute of Bioorganic Chemistry, Russian Academy of Sciences, 117997 Moscow, Russia; 7Belozersky Institute of Physico-Chemical Biology, Lomonosov Moscow State University, 119992 Moscow, Russia

**Keywords:** nanomedicine, targeted therapies, EPR, SPION, Abraxane, Doxil, micelles, exosomes

## Abstract

The ultimate goal of nanomedicine has always been the generation of translational technologies that can ameliorate current therapies. Cancer disease represented the primary target of nanotechnology applied to medicine, since its clinical management is characterized by very toxic therapeutics. In this effort, nanomedicine showed the potential to improve the targeting of different drugs by improving their pharmacokinetics properties and to provide the means to generate new concept of treatments based on physical treatments and biologics. In this review, we considered different platforms that reached the clinical trial investigation, providing an objective analysis about their physical and chemical properties and the working mechanism at the basis of their tumoritr opic properties. With this review, we aim to help other scientists in the field in conceiving their delivering platforms for clinical translation by providing solid examples of technologies that eventually were tested and sometimes approved for human therapy.

## 1. Introduction

Nanotechnology development for the medical field has always been focused primarily on translational purposes [1]. Nanomedicine was conceived to increase the safety of very toxic drugs, providing the therapeutics a means for targeting the sick tissues [2]. For this reason, nanotherapeutics were tested extensively to improve chemotherapy performances. Most of the investigations, in fact, were dedicated to enhance the antitumor power of drugs that were already approved in clinics, but characterized by severe adverse effects limiting their use [3]. Liposomal formulation of Doxorubicin (DOX) [4] and Daunorubicin [5] were the first FDA approved nanotherapies in 1995 and 1996, respectively. First generation of nanoparticles targeted the cancer lesions by exploiting leaky tumor vasculature and the lack of an efficient tumor lymphatic system, a phenomenon known as enhanced permeability and retention effect (EPR) [6]. In this case, the nanoparticles were designed with high circulation properties to facilitate their accumulation in the tumor tissue, where they could release their therapeutic payload. Here, surface modifications like polyethylene glycol (PEG) could minimize particle sequestration in the elements of the mononuclear phagocytic system (MPS), inhibiting particle opsonization and internalization [7]. Second generation of nanomedicine provided the carriers with more complex surface modifications [8] or particular shapes [9] that alone or in combination with “stealthing” molecules allowed for active targeting. Most of these modifications were based on peptides, antibodies or ligand moieties that could be recognized by surface receptors over-expressed on cancer cells [10]. This strategy allowed for a more stable interaction of the NPs with cancer cells and favored their internalization [10]. In these technologies, cancer lesion targeting still occurs via EPR, even though some surface modification can impart the carriers with trafficking properties that can enhance their active accumulation in the sick tissue. A recent review by Anselmo et al. [11], showed that in the last 3 decades, a little more 30 than nanoformulations were approved and less than half of them were designed for cancer treatment. The translation of nanomedicine to the clinic has been hindered by the formidable ability of our body to recognize foreign bodies [12], tumor organization [13], and concerns derived from systemic toxicity and immune system activation [3]. Further efforts in facing this issue generated more and more complex surface modifications, as well as carriers derived from natural sources like exosomes [14]. Unfortunately, while in preclinical settings these technologies provided promising results, their clinical translation was hampered by their high costs of production and sometimes by issues of generating nanoformulations in large scale [15,16]. However, the research in the field is still very active and supported by the exploration of alternative administration routes other than intravenous [17,18] and by the development of other applications, including diagnosis. The goal of this review is to summarize the nanocarriers that reached clinical trial experimentation and some examples to the readers about the characteristics that nanoformulations should have to treat the patients, including their size and targeting mechanism. In this goal, we selected the clinical trials on clinicaltrials.gov by searching for “cancer disease” and “nanoparticles”. We excluded suspended, terminated and withdrawn trials. An additional filter was applied to exclude clinical trials that did not investigate nanoformulations for their ability to treat cancer conditions. Considering that some of these studies were performed to test the same technology against different cancers or, in combination with different therapeutics, our work will focus on no more than 15–20 different delivery systems.

## 2. Inorganic NPs 

### 2.1. AGuIX

A promising approach to overcome current therapies limitations is the use of new therapeutic agents (molecules or nanoparticles) that sensitize cancer cells to radiotherapy (RT) also known as dose enhancers and radiosensitizers. This enhanced radiation local absorption in combination with the accumulation of high Z-elements in the irradiated cancer tissue results in a larger production of harmful diffused photons, photoelectrons, Auger electrons, Compton electrons, and radical species [19]. AGuIX particles were firstly synthesized in 2011 [20] to increase the radiobiological effect of high-energy radiation in the tumor. This technology is composed of very small (average size 5 nm diameter) polysyloxane particles with chelated cyclic gadolinium covalently grafted into the inorganic matrix [21] (Figure 1). In preclinical experiments, AGuIX NPs showed high radiosensitizing and anti-tumor properties [22] that combined with a solid and reproducible synthesis process favored their evaluation in clinical settings.

AGuIX biodistribution studies have been performed on healthy animals of different species (rodents and monkeys) and using different administration methods, doses, and time windows of detection biodistribution and PK properties of these particles [21]. After intravenous administration, carriers’ excretion occurred mainly through the kidneys and only less than 0.15% of the administered dose was found in organs other than the kidneys and bladder [20]. These data were confirmed also by another study [23], showing that kidney accumulation already occurred 5 min after the particle administration, with the highest accumulation observed after 4 h, while the gadolinium clearance occurred in a week. 

AGuIX have been tested on various central nervous system tumors. These investigations demonstrated that tumor accumulation occurred via EPR, indicated as the mechanism of particle tumor targeting [24], while retention in the sick tissue persisted up to 24 h after intravenous administration [25]. In addition, also intraperitoneal administration was effective in targeting the tumors, without evident accumulation in off-target organs besides the kidneys and bladder. AGuIX showed radiosensitizing efficacy in vitro on various cell lines with typical sensitizer enhancement ratios varying from 1.1 to 2.5 for photon irradiation at different energies ranging from keV to MeV, including the use of clinical irradiators [26,27,28].

The first clinical trial NANO-RAD, investigating the AGuIX benefits [29], was completed in 2019 and aimed at treating multiple brain metastases. This trial was designed to determine the maximum tolerated dose in combination with whole brain radiation therapy (RT). The efficiency of AGuIX in overcoming the blood—brain barrier in brain metastases, was similar to conventional magnetic resonance imaging (MRI) contrast agents. Good tolerability of AGuIX intravenous injection was shown at doses ≤ 100 mg/kg in combination with whole brain RT in patients with multiple brain metastases [30]. Most of the ongoing clinical trials (Phase 1/2) are in the recruitment state and focusing on brain tumor metastases or glioma [31,32,33]. Other clinical trial investigate AGuIX in combination with cisplatin, radiotherapy and brachytherapy in treating advanced cervical cancer [34] in combination with MRI-guided stereotactic body RT in pancreatic and lung tumor treatment [35]. 

### 2.2. NBTXR3

NBTXR3 technology is a novel radio sensitizer comprising crystalized hafnium oxide (HfO2) nanoparticles, locally injected into tumor tissue and activated by RT. HfO2 nanoparticles possess excellent x-ray absorption coefficient because of the high electron-density elements composing the particles and acceptable safety. The particles are 50 nm in size and negatively charged thanks to a phosphate coating applied to maintain colloidal stability [36]. NBTXR3 followed by RT could improve the treatment of advanced or borderline-resectable cancers compared to RT alone [36]. Preclinical studies have shown that NBTXR3 working mechanism is mostly physical without targeting specific biological pathways, and its use could be extended to many types of cancer. The system was tested in patients with head and neck squamous cell carcinoma exploring a dose escalation setting [37]. Within 7 weeks after NBTXR3 injection, nanoparticles in the surrounding tissues disappeared, showing that the system was well tolerated. Additionally, this study [37] showed that one intratumor administration of NBTXR3 before radiotherapy could yield remarkable local tolerance, homogeneous dispersion of the particles in the tumor tissue, no leakage, and showed promising signs of anticancer activity in terms of pathological responses. In another study [38] dose optimization and side effects were evaluated for NBTXR3 in association with RT for recurrent and inoperable non-small cell lung cancer patients.

The purpose of NCT04484909 [39] and NCT04615013 [40] trials (phase I) was to determine the recommended phase 2 dose and safety profile of NBTXR3 activated by radiation therapy to treat metastatic, borderline-resectable pancreatic cancers and esophageal adenocarcinoma, respectively. 

NBTXR3 nanoparticles have become the subject of many clinical trials to treat solid tumor with metastases to lung and/or liver [41], head and neck cancer [42,43] and soft tissue sarcoma [44]. Here, the patients underwent radiotherapy with or without a previous local injection of the radiosensitizer. The presence NBTXR3 doubled the pathological complete response of the patients with no occurrence of important adverse effect [45]. Promising results were collected also in a Phase 1 trial investigating the safety and efficacy of this technology in elderly patient affected by oropharynx and oral cavity cancer [46].

### 2.3. Super Magnetic Iron Oxide 

Superparamagnetic iron oxide nanoparticles (SPION) found their application in the biomedical field because of their theranostic properties, since they can allow for MRI and thermo-ablation. SPION working mechanism depends on an external alternate magnetic field determining their action only in the sick tissue. The synthesis of these particles is based on the nanomanipulation of magnetite and maghemite [47] and their dispersion is obtained through surface modifications (capping) based on organic molecules [48] and polymeric (i.e., polyethylene glycol) [49] surface functionalization. Their size ranges from 20 to 150 nm [47] and the mechanism of tumor accumulation is based on intratumoral injection [50] or EPR [51] following IV administration. However, in preclinical studies they were object of intense studies to engineering their surface with targeting molecules [52] to increase their residence time and internalization in cancer cells. These manipulations focused also on conjugating therapeutic agents on their surface comprising both small molecules [53] and biologics [50]. Attempts at exploiting polarized magnetic fields to increase their tumor targeting were proposed as well [54]. Finally, they were often used to implement the properties of other delivery platforms in hybrid synthetic settings [55]. Their translational use was deeply investigated mostly for their ability to enhance the MRI resolution, even though their toxicity related to DNA damage and reactive oxygen species formation limited their large application [56]. FDA-approved formulations of SPION are currently intended as iron replacement and they are secondarily used as contrast agents for kidney imaging [57,58]. In this scenario, the only phase 3 and 4 trial focused on this technology aimed at understanding the SPION ability to detect lymphatic metastases in breast [59] and pancreatic cancer [60] after IV infusion. The latter trial showed, in comparison with traditional histology, a matching of the 2 methods higher than the 80%. Ongoing clinical trials aim to investigate the efficacy of locally injected SPION magnetic hyperthermia against brain and prostate tumors. In a phase I clinical trial, the authors showed the effect of thermoablation to treat prostate cancer by local injection of SPION and further thermoablation induction under magnetic field application [61]. Untargeted SPION (Ferumoxytol) are currently evaluated for treatment of primary and metastatic hepatic cancers [62]. The radiotherapy with SPION supported by magnetic resonance imaging guided linear accelerator allowed to detect and maximize avoidance of residual functionally active hepatic parenchyma from over-the-threshold irradiation thus minimizing SBRT liver damages because of stereotactic body radiation therapy in patients with pre-existing hepatic conditions. The safety, efficacy and tolerability of SPION in combination with spinning magnetic field (SMF) and neoadjuvant chemotherapy in osteosarcoma patients is currently evaluated in a Phase I clinical trial [63]. The study comprises intra-tumor injection of SPIONs followed by SMF and conventional neoadjuvant chemotherapy from day 1. The authors declare a synergistic effect of SPIONs/SMF with neoadjuvant chemotherapy in increasing cancer cell killing and improving the ratio of limb retention (amputation). 

### 2.4. Gold Nanoparticles

Besides their use for increasing the sensitivity of current diagnostic and prognostic tests [64], NU-0129 gold nanoparticles were tested in a Phase 0 clinical trial against gliosarcoma and glioblastoma [65,66]. The particles were modified covalently with a spherical RNAi corona targeting BCL2L12 messenger. The author could detect the accumulation of the particles in the tumor and a decrease in the BCL2L12 expression. However, the authors of this work could not find more information of this technology in terms of size and surface charge, speculating that the ability of this technology to overcome the blood–brain barrier is probably due to their small size, as demonstrated in other pre-clinical studies [67]. On the other hand, the overcoming of the blood–brain barrier could be favored by the nucleic acid coating of the particles since spherical RNAi could be trafficked via transcytosis, with the gold core fundamental to avoid the fast clearance of the system [66]. 

### 2.5. ELU001 (Folic-Acid Functionalized C’Dot-Drug-Conjugate)

The folate receptor alpha (FRα) represents a promising target in oncology because of its over-expression in tumors (i.e., ovarian, breast and lung cancers), low and restricted distribution in normal tissues [68], emerging insights about its tumor promoting functions, and association with patient prognosis. ELU001 is a new molecular C’Dot Drug Conjugate (CDC). ELU001 comprises a very small silica core (6 nm) functionalized with ~12 folic acid targeting moieties and ~22 exatecan topoisomerase-1 inhibitor payloads linked to via Cathepsin-B cleavable linkers covalently bound to the surface of the nanoparticles. Because of their small size, ELU001 are characterized by tumor penetration ability via receptor-mediated endocytosis and are rapidly eliminated by the kidneys. ELU001 high avidity is believed to promote internalization into FRα over-expressing cancer cells, selectively delivering its therapeutic payload. The first Phase I/II clinical trial [69] dedicated to ELU-FRα-1 is under recruiting phase for advanced, recurrent or refractory FRα over-expressing tumors, considered being topoisomerase 1 inhibitor-sensitive [70] and with no other therapeutic options available. The study will focus on dose escalation and safety to determine the recommended Phase 2 dose and on the expansion of the patient cohort, where specific cancer types will be evaluated for efficacy and safety of this technology.

## 3. Polymeric Particles

### 3.1. CALAA-01

CALAA-01 is considered the first targeted polymeric carrier designed for delivering siRNA tested in human [71]. A positively charged cyclodextrin core allows the loading the negative nucleic acid payload. Polydispersity, stability, circulation, and targeting properties depend on surface PEGylation and transferrin modification (Figure 2) [72]. 

The use of transferrin as targeting agent is very common in nanomedicine applied to cancer disease, since many tumors over-express the receptor for this molecule [73]. When fully assembled, the carriers have an average size of 50–70 nm and can be trafficked in the endosomal compartment of the cells after internalization favored by the interaction with transferrin receptor, even though EPR is still fundamental for the particles to reach cancer cells [71]. After sequestration in the endosomal compartment, the system can disassemble and eventually escape from these vesicles, probably thanks to the positive charge of the cyclodextrin structure. The system was enriched with imidazole groups to favor the buffering properties of the system and induce particle endosomal escape via proton sponge effect [71]. In a Phase 1 clinical trial [74], the system was tested for its efficacy and safety against solid tumors (i.e., melanoma) by delivering siRNA against the M2 subunit of ribonucleotide reductase. The trial was terminated before being completed, perhaps because of toxicity issues probably related to the carrier [75], and the results were not reported. However, a work on this trial was published showing the ability of this technology to induce tumor regression and decreasing the expression of functional ribonucleotide reductase [76].

### 3.2. Micelles

Polymeric micelles are nanocarriers composed of a core–shell structure that can be generated via self-assembly of amphiphilic block copolymers [77]. Because of their self-assembly and amphiphilic nature, micelles are relatively easy to synthesize compared to other technologies and for this reason they are often studied as drug delivery vehicles for poor water-soluble compounds [78]. Hydrophilic polymers including (but not limited to) PEG, polyoxazolines, chitosan, dextran, and hyaluronic acids can wrap their hydrophobic core, while the therapeutic payloads can be also chemically conjugated to these structures [79]. Micelle surface can be easily modified in function of the number of monomers used in their fabrication and conjugation of tumor-specific ligands is easy and reproducible [79,80]. Regarding their clinical translation, one of the major limitations of micelles is represented by their low mechanical properties and re-assembly when their amount in aqueous solution is below the so-called critical micellar concentration [77]. Currently, there are several micellar-based nanoformulations approved for improving cancer treatment, and others are in advanced clinical trials. The chemotherapeutic drug Paclitaxel (PTX) that has a very low solubility in water (less than 0.1 µg/mL) is often used in micellar-based systems to avoid Cremophor-EL and ethanol formulations, resulting in adverse reactions like dyspnea, hypotension, angioedema, and generalized hives (2–4% of patients). Genexol-PM, a monomethoxy-poly (ethylene glycol)-block-poly(D,L-lactide) with a mean size of 20–50 nm was approved in clinics in several Asian countries (South Korea, Philippines, India, and Vietnam) for breast cancer, lung cancer, and ovarian cancer [78]. In Genexol-PM, PTX is physically incorporated into the inner core of the micelles that target the tumors via EPR effect. Currently, Genexol-PM in combination with carboplatin is tested for its safety as an adjuvant treatment in patients with newly diagnosed ovarian cancer that underwent cytoreductive surgery [81]. Other polymeric micelles represent a promising vehicle for PTX delivery, and they show similarity with Genexol-PM including a core–shell structure with physical entrapment of PTX, PEG coating, small size, and passive targeting through the EPR mechanism. Despite there are limited evidence of superior efficacy of polymeric-PTX compared to Cremophor-PTX, micelles allow administration of an increased PTX dose and offer improved patient safety. Similar platforms worth to mention are Apalea/Paclical (mean size 20–30 nm) and pm-Pac (mean size 20 nm) that target the tumor via EPR [82,83]. Apalea/Paclical contains retinoic acid to solubilize PTX and is approved in different countries (Russian Federation, Kazakhstan, and European Union) against platinum-sensitive ovarian, peritoneal, and fallopian tube cancer [84,85]. Finally, pm-Pac (mean size 20 nm) successfully passed a Phase III study as first-line treatment in combination with cisplatin for advanced non-small cell lung cancer (NSCLC) [86,87]. Trials (Phase 1–3) dedicated to investigate the benefits of micellar-based technologies are currently ongoing in China [88] and Japan [89] where they showed similar therapeutic benefits compared to PTX, but less toxicity in treating metastatic or recurrent breast cancer. 

Docetaxel (DTX) is another taxane with solubility issues. An analog of Genexol-PM, called Nanoxel-PM micelles loaded with DTX [90] is currently under clinical trial for efficacy evaluation against different cancers [91]. Additionally, it is tested as neoadjuvant in patients with breast cancer in combination with DOX and cyclophosphamide [92] and against salivary duct carcinoma in combination with anti-HER2 monoclonal antibody [93]. Other versions of this therapeutic formulation are tested in trials as well [93]. Micelle-based technologies are under clinical trial also for evaluating their ability to deliver cisplatin. NC-6004 micelles have an average size of 30 nm and are composed by PEG and poly-glutamic acid copolymers (PGlu). NC-6004 combination with gemcitabine (GEM) has been studied in NSCLC patients, biliary tract, and bladder cancer patients [94] resulting in long-lasting antitumor activity and favorable safety profile. Similar data were registered in combination with Pembrolizumab in the treatment of head and neck cancer [95] and in combination with GEM against advanced solid tumors [96]. Similar formulations are widely investigated [97] including the NK012 where the payload SN-38 is covalently attached to the PGlu structure. Here, the efficacy of NK012 was tested in patients with not-resectable colon cancer, but more data are necessary to evaluate its benefits in comparison with the common treatment irinotecan [98,99]. A novel epirubicin drug conjugated polymeric micelle (NC-6300; 40–80 nm in diameter) was developed by conjugating the payload to PEG polyaspartate block copolymer through a pH-sensitive linker which enables the selective epirubicin release in tumor. This technology exploits tumor pH as targeting, representing a perfect example of smart technology in clinics in the treatment of cutaneous and not cutaneous angiosarcoma [100] and advanced, metastatic, or unresectable solid tumors, including soft-tissue sarcomas [101].

### 3.3. EP0057

EP0057 (formerly known as CRLX101) is a formulation of camptothecin (CPT) conjugated with a cyclodextrin polymer backbone and is currently being evaluated clinically in multiple refractory solid tumors [102,103,104,105,106]. The micelles have a size of approximately 30–40 nm and significantly increase CPT (topoisomerase I inhibitor) solubility while preserving its active lactone form [107]. EP0057 also exhibits better patient tolerance than other CPT analogs. The nanoparticles-drug conjugate is administered via intravenous infusions, and nanoparticles preferentially accumulate in the tumors through EPR [107]. EP0057 has been shown to inhibit significantly also hypoxia-inducible factor-1 alpha (HIF-1α) and therefore serving as a radiosensitizer with the potential to improve the efficacy of chemoradiation therapy [108,109,110]. 

Prior studies showed EP0057 efficacy in recurrent or persistent, epithelial ovarian, fallopian tube or colorectal, peritoneal, and gastroesophageal cancer [111,112], where it showed promising results [112]. Ongoing clinical trials are designed to evaluate the efficacy and safety (Phase 1/2) of this therapeutic in lung [102], gastric [113] and ovarian [103] cancer in combination with Olaparib, as well as to evaluate its pharmacokinetics properties (PK) [102,114] using a population model. From the data obtained from 27 patients enrolled on two-Phase II clinical trials, the release of CPT was characterized by an initial rapid clearance, which decreased via first-order decay to the steady-state value by 4 h after the infusion. A second Phase I/IIa clinical study involved 22 efficacy-evaluable patients with metastatic renal cell carcinoma, who received increasing doses of EP0057 combined with bevacizumab [106]. Partial response or stable disease was observed in 86% of the patients, with a median progression free survival (mPFS) of 9.9 months. Most patients achieved a reduction of tumor and increased the progression-free survival compared to their previous therapy [115]. In addition, a Phase Ib/II study of EP0057 combined with PTX in women with recurrent epithelial ovarian cancer reported a 31.6% overall response rate, including one complete response with a 5.4 month median progression-free survival [105]. However, the analysis of trials including data highlights the need of more investigation to evaluate the clinical benefits of this therapeutic also because of the onset of considerable side effects [116,117,118,119]. 

### 3.4. NanoPac

Other attempts to formulate PTX in nanostructure to avoid the use of Cremophore-EL were performed. NanoPac (also known as, Nanotax) are pure PTX nanoparticles generated in supercritical carbon oxide environment in the presence of organic solvents. These particles have a size of 600–800 nm inhibiting their clearance and making their use helpful for topical and local administration (i.e., inhalation) [120], with no targeting mechanism associated. A Phase 2 trial focused on investigating the effects of different concentration of NanoPac against prostate cancer directly injected into the prostate. Interestingly, the lower dose of drug showed higher benefits in terms of tumor reduction compared to higher doses. The drug showed reasonable side effects, also at the highest dose used [121]. In similar experimental settings, other trials measured the ability of intra-cystic injected Nanopac to contrast the progress of pancreatic cancer [122] and of intraperitoneal administration against ovarian cancer [123]. Additionally, in these cases, lower doses of NanoPac showed higher clinical benefits even though the occurrence of side effect was significantly more pronounced.

## 4. Abraxane and Related Technologies

### 4.1. Abraxane

ABI 007, nanoparticle albumin-bound (Nab) PTX, Abraxane are all names used to identify albumin nanocarriers loaded with PTX, to avoid the use of Cremophor-EL [124]. Abraxane allowed for PTX encapsulation, exploiting the natural ability of Albumin to interact with hydrophobic drugs at multiple sites of its structure [124]. The interactions between albumin and drug are solely hydrophobic without formation of covalent bounds, even though some low-degree of crosslinking that can occur between the albumin molecules on the surface of the nanoparticles [124]. The nanocomplexes have an average size diameter of 130 nm and the energy to generate the hydrophobic interactions is provided by the synthetic route, based on high-pressure homogenization where drug and albumin molecules are mixed in an aqueous solution and pushed (under high pressure) in the narrow spaces of the homogenizer (Figure 3) [125]. 

Besides the generation of a safer PTX formulation, Abraxane demonstrated also significant benefits in terms of PK, fast drug distribution and an increased distribution volume. While the size of these particles allowed for exploiting the EPR, it is important to highlight that the nanocomplex degradation in circulation is quick and can lead to the formation of single molecules of albumin bound to PTX. In this scenario, the trafficking of albumin can be also regulated by endothelial receptors like GP60 that favor caveolae-mediated translocation of albumin from the lumen of the blood vessels to the sub-endothelial space. In a trial focusing on NSCLC treatment [126,127], a correlation between the expression of caveolin-1 and the drug efficacy was registered, showing this transporter as one of the mechanism of Abraxane tumor trafficking [126]. 

Here, some authors show that secreted protein acid rich in cysteine (SPARC), over-expressed on the surface of different cancer cells including pancreatic cancer [128] might eventually favor Abraxane internalization. Additionally, SPARC [129] was object of clinical investigations in pancreatic cancer. These studies confirmed a correlation between Abraxane and SPARC expression [130,131,132]. However, it is worth to mention that Abraxane trafficking was never proved directly and that recent evidence shows other receptors evolved to manage the trafficking of denaturated albumin could be responsible for Abraxane trafficking to the tumor [133]. Abraxane was approved by FDA and EMA for metastatic breast cancer, locally advanced or metastatic NSCLC [124], and as the first-line treatment of metastatic adenocarcinoma of the pancreas [134]. Phase 3 and 4 clinical trials demonstrated Abraxane is more effective when combined with other drugs like atezolizumab [135], GEM and carboplatin in contrasting triple negative breast cancer [136], with GEM against pancreatic cancer [137] and melanoma [138], and with carboplatin against NSCLC [139]. On the other hand, with bevacizumab (and sometimes other agents) the onset of serious side effects was registered [140,141,142,143,144,145], even though slightly beneficial effects on tumor growth were observed like in patients with inoperable melanoma [146,147]. Abraxane was shown also to improve the effect of biological therapy like the immune modulator TLR-7 activator Imiquimod [148,149]. Another clinical trial showed that the best effect of Abraxane as single agent for treating metastatic breast cancer is achieved by prolonging the administration time of reduced doses, after a brief period at normal doses administration [150]. Abraxane showed beneficial effects also on pancreatic cancer as secondary line therapy in patients that progressed on GEM [129] and it was tested also to reduce tumor mass before radiotherapy and surgery in combination with GEM [151], even though its efficacy did not peak significantly higher compared to irinotecan/oxaliplatin/5-fluorouracyl combination [130,131,132]. Finally, it is worth it to mention that Abraxane was tested also against non-Hodgkin’s lymphoma [152]. Here, the system was covalently coated with rituximab, generating a new carrier with the toxicity of Abraxane and the targeting/toxicity of Rituximab against CD20 positive cancer cells [153]. 

### 4.2. Abi 009 

The same technology at the basis of Abraxane was applied to deliver Rapamycin. Nab-Rapamycin or Abi 009 is a colloidal albumin nanoformulation loaded with this therapeutic that affects cancer viability via mTOR inhibition. Additionally, in this case, encapsulation in albumin complexes resulted very useful to deliver very hydrophobic molecules and it is normal to expect that the same approach could be used to deliver other drugs with the same physical and chemical features as well as targeting mechanism. The only completed trials by far were performed against advanced carcinomas characterized by mTOR mutations [154] and to evaluate the effective not toxic doses (Phase 1/2) of AB009 in the treatment of bladder cancer. Here, the particles were administered directly in the bladder at different concentrations, and retention time in the organ and with and without GEM. All the conditions tested showed good tolerability [155]. Other active or recruiting trials are testing the ability of ABI009 (alone or in combination with other drugs) against sarcoma [156], and different solid pediatric tumors including the central nervous system cancer [157]. In this scenario also albumin nanoparticle loaded with sirolimus (mTOR inhibitor) was tested against glioblastoma [158]. 

## 5. Lipid/Proteolipid Technologies

### 5.1. Doxil

Doxil (also known as Caelix) is a PEGylated liposomal formulation of DOX (average size 70–100 nm), a molecule that intercalates into nucleic acids, inhibiting topoisomerase II enzyme, DNA/RNA synthesis and causing oxidative damage to nucleic acids, proteins and lipids [4]. In this multi-component system, DOX is loaded into the liposome core, providing efficient drug targeting via EPR effect. The system is PEGylated to prolong its circulation time and protect it from MPS sequestration. Interesting, the intellectual property of the system is not based on the liposomal formulation, but on the drug loading mechanism based on a transmembrane ammonium sulfate gradient. Ammonia, continuously produced by tumor cells during glutaminolysis, allows the efficient drug release at the site of the tumor [159]. Doxil was the first nanotherapeutic approved to treat cancer, and it is used for treating ovarian cancer, AIDS-related Kaposi’s sarcoma, and multiple myeloma [160]. It is also the first nanotherapeutic approved in the generic version under the name of Lipodox [161].

While Doxil has been widely used in the clinic since 1995, its efficacy was variable in different cancers. In some pathologies, such as metastatic breast cancer, Doxil efficacy was not significantly different from free DOX [162]. This controversy was attributed to the limited or insignificant role of EPR in many “cold” human cancers [163], blunting the anti-cancer efficacy of the drug because of liposome-mediated activation of macrophages enhancing tumor growth, and suboptimal release of the drug from liposomes in the tumor bed. 

Currently, no new clinical trials of Doxil have been started. The most recent trials include 5 Phase III/Phase IV studies between 2008 and 2019, two of them were completed, and three terminated. These trials include the investigation of Doxil for treatment of newly diagnosed multiple myeloma [164], advanced and/or metastatic breast cancer [165,166], recurrent epithelial ovarian carcinoma [167], and advanced-relapsed epithelial ovarian, primary peritoneal, or fallopian tube cancer [168]. 

In a Phase IV trial to study Doxil as a monotherapy for 25 patients with locally advanced and/or metastatic breast cancer [165], Doxil was administered to elderly women (>65 years old) every 28 days until treatment failure. As a result, 16% of patients had to discontinue treatment because of adverse effects (cardiac events and palmar–plantar-erythrodysestesia), 14% had a partial response, no patients had a complete response, and the rest (60%) had stable disease by the end of the study. Overall survival of patients receiving Doxil was 20.6 months, and time to progression was 5.7 months [163]. Based on these results, the authors concluded Doxil is a safe and effective in elderly breast cancer patients. 

The most recent trial (results posted in 2019) analyzed the combination of trabectedin with Doxil for treating recurrent ovarian cancer [168]. The aim of the trial was to compare the overall survival of women with platinum-sensitive, recurrent ovarian cancer treated with Doxil as a monotherapy or in combination with third-line drug trabectedin. It was found that none of the patients reached primary endpoints of overall survival, and the combination therapy did not show any advantage compared to monotherapy in terms of overall survival or safety. However, a subset of patients bearing a BRCA1/2 mutation in the combination therapy group showed a clinically significant reduction in the risk of death (45.8%) [169]. The effect of BRCA1/2 mutations is consistent with previous reports which found that Doxil plus trabectedin therapy prolonged patient overall survival compared to Doxil monotherapy [170]. In patients with advanced soft tissue sarcomas and recurrent ovarian cancer, it was found that trabectedin and Doxil combination showed increased risk of cardiac-related treatment-emergent adverse effects [171], and adverse effects emerged also in other trials [166,167] where Doxil safety was lower than expected (i.e., when used in combination with [164] thalidomide and dexamethasone in patients with newly diagnosed multiple myeloma). 

### 5.2. Other Liposomal Formulations

Onivyde is the nanoformulation of irinotacan. It comprised pegylated liposomes of 110 nm and is approved to treat metastatic adenocarcinoma of the pancreas [172]. The system showed a very high encapsulation yield because of a refined synthetic method based on drug-stabilizing agents like polyphosphate or sucrose octasulfate [173]. Interestingly, it was shown that the interaction between the PEG and the serum protein could increase particle cancer cell internalization, providing a natural-occurring targeting at the level of the cancer lesions, after liposome extravasation via EPR [174]. Besides pancreatic cancer, Onivyde was tested in clinical trial also for lung [175], glioblastoma (administered via convection) [176], oesophagea l [177], and other solid tumors [178].

Myocet is another liposomal formulation of DOX approved to treat breast cancer. Similar to Doxil, the system allowed for increasing DOX efficacy and safety, particularly for what concerns cardiotoxicity. At the basis of the generation of this technology stands a pH gradient that favor DOX encapsulation while drug stabilization in the liposomes is achieved through citrate complexation [179]. This phenomenon is fundamental to avoid particle leakage that occurs preferentially at the cancer lesion because of the presence of high concentrations of phospholipases that induce particle degradation and consequent payload release. The particles have a size of around 150 nm and a loading yield of about 95% [179]. Even though Myocet does not contain any surface modification to prolong its circulation time, the system had enormous benefits in terms of PK [180] increasing the area under the curve of 20 times and targeting the tumor via EPR. Despite clinical trials to contrast metastatic breast cancer, Myocet was tested also against glioma in children [181], lymphomas [182,183], ovarian, fallopian and peritoneal cavity cancer [184,185,186], but to our knowledge the results of these trials were not reported. 

Daunoxome is the liposomal formulation of Daunorubicin and it was approved to treat HIV-related Kaposi’s sarcoma [5]. These particles are about 45 nm in size and even though they are not PEGylated as Doxil, they were designed with a neutral surface charge to minimize opsonization and sequestration by the element of the mononuclear phagocytic system [187]. The particles are intravenously administered and accumulate in the tumor via EPR. The benefits of this formulation were shown in preclinical studies showing an increase of tumor targeting of almost 10 times compared to free-administered drug and a controlled released in the cancer lesion up to 36 h [188]. Daunoxom showed also to protect the drug from biotransformation in toxic and inactive derivates [189]. Besides Kaposi’s sarcoma [5,190], Daunoxome (in combination with other drugs like Cytarabine) was extensively clinically tested to treat acute myeloid leukemia, where it showed higher efficacy compared to free-administered drug and acceptable side effects [189,191].

Vyxeos is the liposomal formulation of Daunorubicin and Cytarabin and it is approved for leukemia treatment [192]. The drugs are loaded with a ratio 1:5 (D/C) because previous reports demonstrated the occurrence of synergistic effects in this formulation. In freeze-dried conditions, the particles have a size of 110 [193] and they are not provided with surface targeting properties or stealthing mechanisms [11]. However, drug encapsulation allowed for better PK properties and when interacting with targeted cells in the bone marrow, the delivery of both the therapeutics allowed for significant improvement in their efficacy, compared to free-administered drugs [194]. In clinics, Vyxeos showed to increase the overall survival of elderly patients with acute myeloid leukemia with side effects comparable to standard treatments [195]. In this regard, a specific trial was dedicated to evaluate kidney toxicity [196].

ATU-027 is a liposomal formulation of about 100 nm in size, designed to deliver siRNA. The system is composed of pegylated, cationic, and fusogenic lipids to allow the particles to load siRNA, maintain proper circulation properties, and deliver the payload in the cell cytoplasm [197]. This liposomal formulation was designed to deliver siRNA against PKN3 expression, a key factor in the downstream pathway of PI3K/PTEN [197]. Here, the liposomes are supposed to target the neo-angiogenesis process, and even though with no specific targeting for endothelial cells, they demonstrated in vitro and in vivo the ability to target angiogenesis and lymphoangiogenesis. The system showed good tolerability in clinical trials against solid tumors [198,199] and in particular against pancreatic cancer in combination with GEM [200,201]. 

Marqibo is a liposomal formulation of vincristine designed to improve the PK properties of this drug and increase its efficacy and safety. The system comprises sphingomyelin/cholesterol liposomes [202] of about 100 nm encapsulated with the drug. The composition of these liposomes is important because it was shown to provide a neutral charge to the particles, decreasing their opsonization and consequent clearance by the MPS. Enhanced PK properties, prolonged circulation and relatively small size can favor tumor targeting via EPR [203]. However, Marqibo was approved to treat Philadelphia chromosome-negative acute lymphoblastic leukemia [204]. The system showed high tolerability also in children receiving adult doses against refractory solid tumors and leukemia [205], while more data are needed to evaluate its efficacy against retinoblastoma [206]. Marqibo showed promising results against metastatic uveal melanoma [207] and B-cell lymphoma in combination with other therapeutics [208].

MEPACT is the liposomal formulation of the immune modulator mifamurtide and it is approved to treat high-grade non-metastatic osteosarcoma in young patients [209]. The liposomes are multilamellar and their size is claimed to be below 100 nm from different sources [172,210], even though other reports claim that this technology is about 3 um in size [211]. Additionally, known as liposomal muramyl tripeptide phosphatidylethanolamine, the system encapsulate a synthetic biologics that can target the macrophages of liver and spleen that in turns can activate other leukocytes against cancer cells [211], decreasing the importance of the size. 

### 5.3. Exosomes

Exosomes are biological extracellular vesicles secreted from different cells with a size of 30–120 nm and a characteristic biological protein identity providing resistance to MPS clearance and tumor tropism (Figure 4) [212]. In the Phase I clinical trial [213] “iExosomes” mesenchymal stem cells (MSC)-derived exosomes loaded with anti-KrasG12D siRNA for treating metastatic pancreatic ductal adenocarcinoma (PDAC) with KrasG12D mutation are tested for the best dose and side effects. Mutations in the GTPase KRAS drive initiation, progression, and metastasis of PDAC. This clinical trial studies the effects of the treatment first described in 2017 by Kamerkar et al. [214] reporting the use of targeted exosomes (iExosomes) loaded with anti-KRAS siRNA and equipped with a CD47 “do not eat me” signal for treating PDAC. iExosomes showed remarkably higher efficiency of siRNA delivery into target pancreatic cells compared to liposomes and non-functionalized exosomes, which was attributed to the functional role of CD47, a signal that helps to evade phagocytosis, in suppressing systemic clearance of nanoparticles. CD47 expression on the surface of iExosomes increased the half-life of nanoparticles in systemic circulation and favored their internalization into pancreatic cells. In preclinical models, iExosomes’ administration resulted in tumor disappearance (an effect that persisted even after 200 days post-treatment). In advanced PDAC models, treated mice responded with partial control of tumor growth, reduced tumor burden, and improved histopathology of the pancreatic tissue. The secondary objectives of this trial include evaluation of iExosomes PK, assessing the disease control rate with the therapy, determining median progression-free survival and overall survival of PDAC patients with the treatment. Overall, this is the first clinical trial on the use of (a) functionalized exosomes; (b) exosomes loaded with siRNA and (c) exosomes with CD47 signal protecting nanoparticles from phagocytosis. In case of success, this study could lay a firm ground for further development of exosome-based therapeutics and exosome nanovehicles for treating cancer and non-cancer diseases. 

In another Phase I clinical trial [215] exosomes were tested to deliver of curcumin against colon cancer. This drug showed anti-inflammatory, antioxidant and antitumor activity, but frustratingly low bioavailability. The strong inhibitory effects of curcumin on many colon cancer cell lines were previously showed [216]. These effects were mainly attributed to signal transduction pathways’ modulation, including β-catenin and NF-κB. This trial attempts to use plant (fruit-derived) exosomes to increase curcumin bioavailability and delivery into colon tumors. In this early clinical trial, the effects of curcumin-loaded exosomes on the immune modulation, phospholipid profile and cellular metabolism of colon cancer and normal colon cells will be analyzed. While the method for loading of curcumin into exosomes is not described, it is most likely related to natural binding of curcumin, as a hydrophobic drug, to these particles. Plant-derived exosomes from different fruits are also well described, as well as their uptake by intestine and immune cells [217]. 

Grape exosomes [218] containing Lortab drug (acetaminophen and hydrocodone, a non-opioid pain reliever) with Fentanyl patch (opioid analgesic) are studied in a trial to evaluate the ability of powdered grape exosomes as anti-inflammatory agents to reduce the incidence of oral mucositis, a frequent adverse effect during radiation and chemotherapy treatment for head and neck tumors. This powder will be administered daily by mouth for 35 days during chemoradiotherapy along with orally prescribed oral mucositis standard therapy (pain relievers and anti-fungal mouth washes). Efficacy will be estimated by pain caused by oral mucositis and levels of immune biomarkers in blood and mucosal tissue. 

Another clinical trial uses dendritic cell-derived exosomes in combination with an immune suppressive drug cyclophosphamide. The study describing this approach was first published in 2016 by Besse et al. [219]. Dendritic cell-derived exosomes (Dex), containing a wide range of antigen presentation, adhesion, costimulatory and docking molecules, were shown to trigger NK and T cell immune responses in Phase I clinical trials [220,221]. In a Phase II trial [222], Dex loaded with MHC class I and class II-restricted cancer antigen showed clinical benefits in patients with inoperable NSCLC including longer progression-free survival in patients with advanced NSCLC. 32% of patients receiving Dex experienced stabilization of cancer growth for over 4 months, but the primary endpoint (to observe 50% of patients with progression-free survival at 4 months after cessation of chemotherapy) was not reached. The authors discussed the obstacles that may have led to this failure, including suboptimal use of cancer-testis antigens loaded into exosomes, the use of IFN-γ during manufacturing of Dex that may upregulate PD-1 ligands on exosomes and potential benefit of using Dex in combination with immune checkpoint blockers in lung cancer. Still, this strategy demonstrated activation of antitumor immunity, feasibility of using dendritic cell-derived exosomes for immunotherapy, and clinical benefits for patients with advanced NSCLC. Using Dex may also be portrayed for other cancers with NKp30-specific functional defects, such as gastrointestinal stromal tumors, neuroblastoma, and chronic lymphocytic leukemia. The different technologies presented in this work are summarized in Table 1.

## 6. Conclusions

In this work, we analyzed different nanotherapeutics that reach clinical trial evaluation excluding nanotherapeutics tested for diagnostic [64,223], improvement of surgical interventions [224,225], pain relievers [226] or cancer vaccines [227,228]. Nanomedicine showed the potential to improve the delivery of small molecules and biologics and to represent a means to develop treatments based on physical methods. Most of these carriers accumulate in the tumor via EPR and a small size (below 150 nm) represents a common factor in the clinical success of nanotherapeutics. Only a few of them were engineered with targeting moieties compatible with high-scale production. In the considered studies emerged that, trafficking complexity could be reached only via biology-inspired carriers like Abraxane and exosomes and, for this reason, more investigations in this direction should be performed [229]. It has also to be noted that most times the nanoformulations that reached clinical trial evaluation were tested for delivering other therapeutics with similar chemical and physical properties, developing novel combinatorial approaches and contrasting the growth of different tumor diseases. In this context, the success obtained against one kind of cancer did not guarantee the same results in other oncological diseases, highlighting the importance to characterize the trafficking properties of each oncological disease. 

## Figures and Tables

**Figure 1 ijms-23-13368-f001:**
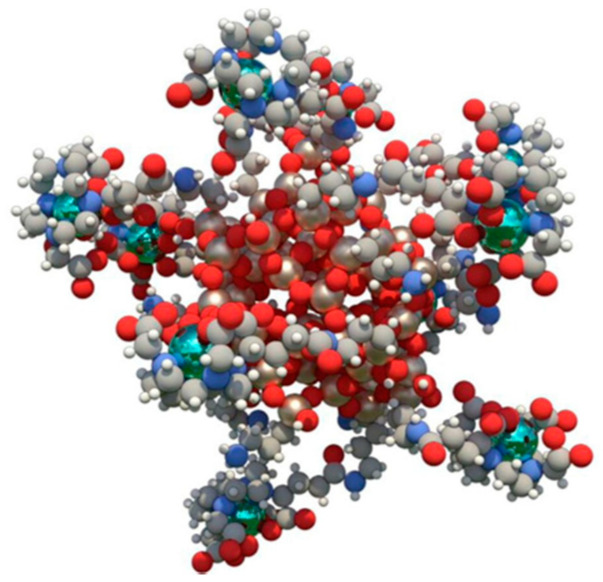
AGuIX structure: Polysiloxane structure of AGuIX chelating gadolinium (green) via dodecane tetraacetic acid ligands. Figure reprinted from Lux et al. [21].

**Figure 2 ijms-23-13368-f002:**
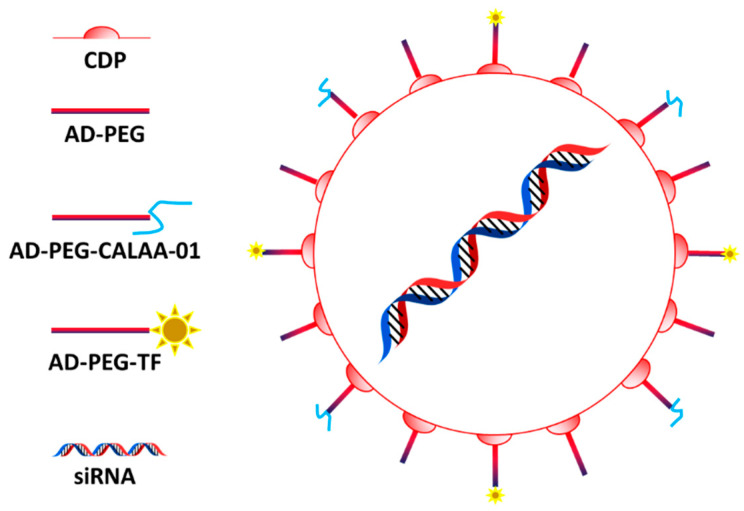
CALAA-01 structure: The system is composed by a core of cyclodestrin encapsulating the siRNA and functionalized with PEG bearing adamantine (AD-PEG) and transferrin (AD-PEG-TF). Figure from Kurreck et al. [72].

**Figure 3 ijms-23-13368-f003:**
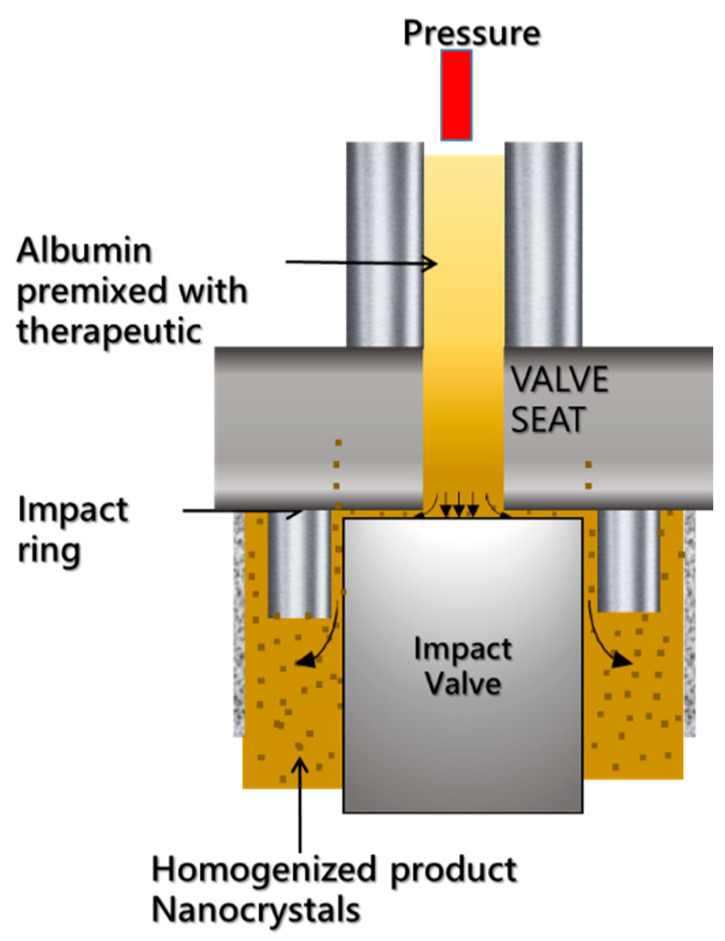
Abraxane synthesis: The particles are generated via high-pressure homogenization where PTX and Albumin interactions occur generating a stable complex. Figure reproduced from Parodi et al. [125].

**Figure 4 ijms-23-13368-f004:**
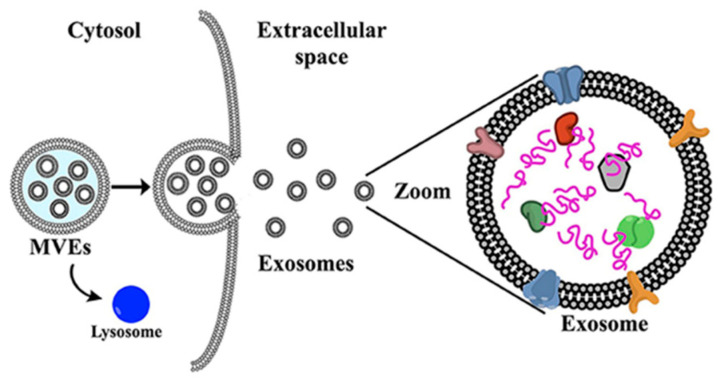
**Exosome origin:** Exosomes derive from the secretion of multivesicular endosome and can be purified and used for drug delivery purposes. Figure reprinted from de la Torre Gomez et al. [212].

**Table 1 ijms-23-13368-t001:** Characteristics and targeting mechanism of nanocarriers in clinical trials.

Particle	Material	Size (nm)	Targeting Mechanism	Killing Mechanism
AguIX	Polysyloxane	5	EPR	radiosensitizer
NBTXR3	Hafnium oxide	50	intratumor	radiosensitizer
SPION	Iron oxide	20–150	EPR, intratumor, polarized magnetic fields	Thermoablation, conjugated drug
NU-0129	Gold	N.A.	EPR, transcytosis	RNAi
ELU001	Silica	6	Folic acid	topoisomerase-1 inhibitor
CALAA-01	Cyclodextrin	50–70	Transferrin	RNAi
Genexol-PM	Polymeric micelles	20–50	EPR	PTX
Apalea/ Paclical	Polymeric micelles	20–30	EPR	PTX
Pm-PAC	Polymeric micelles	~20	EPR	PTX
Nanoxel-PM	Polymeric micelles	10–50	EPR	DTX
NC-6004	Polymeric micelles	~30	EPR	GEM
NK012	Polymeric micelles	~20	EPR	SN-30
NC-6300	Polymeric micelles	40–80	EPR, acidic pH	epirubicin
EP0057	Polymeric micelles	30–40	EPR	Camptothecin, radiosensitizer
Abraxane	Albumin	130	Receptor mediated trafficking	PTX
Abi-009	Albumin	130	Receptor mediated trafficking local administration	Rapamycin
NanoPac	PTX	600–800	Local administration	PTX
Doxil	Lipids	70–100	EPR	DOX
Myocet	Lipids	150	EPR	DOX
Marqibo	Lipids	100	EPR	Vincristine
MEPACT	Lipids	>100	Targeting MPS	Mifamurtide
Onyvide	Lipids	110	EPR	Irinotecan
Daunoxome	Lipids	45	EPR	Daunorubicin
Vyxeos	Lipids			Daunorubicin/Cytarabine
ATU-027	Lipids	100	EPR	RNAi
Exosomes	Proteo/lipids	30–120	Natural tropism	RNAi

## Data Availability

Not applicable.

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
