# Peer review of "Anticancer Nanotherapeutics in Clinical Trials: The Work behind Clinical Translation of Nanomedicine"

_ijms, 2022, doi:10.3390/ijms232113368_

Round 1

Reviewer 1 Report

The authors exploit the database clinicaltrials.gov from the US government to present an analysis about the physical and chemical properties of nanotherapeutics and the working mechanism at the basis of their tumoritropic properties.

The work is useful and should be published. It is a bit long but as review paper this can be accepted.

However, I find it a bit strange that the first author, having spent the years from 2010 to 2016 at the Houston Methodist Research Institute where much action was and is going on in nanotherapeutics, does not mention anything in the introduction and references about researchers (except [10]) of this Institution which is a leading research institutions in the USA in this field. After all, the first author learned his trade there. Some references should be added before publication.

The Authors contribution are missing in the paper: the number of co-authors is not negligible.

Finally, the text should be re-read with care. I found several typos, of which some are indicated below with the line number.

Also, its is tested as neoadjuvant in patients with 311

Abraxane was also shown also to im- 434

Doxil was the first nanatherapeutic approved to treat cancer, and it is used for treating 477

occurrence of synrgistic effects in this formulation. In freeze dried 558

against of colon cancer. This drug showed anti-inflammatory, antioxidant and antitumor

633

many colon cancer cell lines in vitro were previously showed [213]. These effects were 635

admiistration of the nanotherapuetic for their use as radiosensitizer and also for their 697

Reviewer 2 Report

This is quite an interesting review article. I have no serious comments, but a few wishes to the authors.

1) Since the authors cite figures from other reviews, however, a literature review was carried out. I recommend the authors to prepare their own figure of the vision of the problem in the vein of the analyzed literature.

2) Since the authors consider the nanomaterials closest to their use in medicine, it would be nice to analyze the received patents and refer them.

3) It is not entirely clear why gold nanoparticles are discussed in the review. First, there are more works on the anticancer effect of gold nanoparticles, including in the form of nanotransporters of active compounds. Why are studies of nanoparticles of silver, cerium, selenium not reflected?

4) The conclusion is very voluminous. Need to reduce

5) The key limitations of the use of nanomaterials should be indicated

Round 2

Reviewer 2 Report

The work can be accepted for publication